# Learning outside the Black-Box:
# The pursuit of interpretable models

**Jonathan Crabbe**
University of Cambridge
jc2133@cam.ac.uk

**Yao Zhang**
University of Cambridge
yz555@cam.ac.uk

**William R. Zame**
University of California Los Angeles
zame@econ.ucla.edu

**Mihaela van der Schaar**
University of Cambridge
University of California Los Angeles
The Alan Turing Institute
mv472@cam.ac.uk

## Abstract

Machine Learning has proved its ability to produce accurate models – but the deployment of these models outside the machine learning community has been hindered by the difficulties of interpreting these models. This paper proposes an algorithm that produces a continuous *global* interpretation of any given continuous black-box function. Our algorithm employs a variation of projection pursuit in which the ridge functions are chosen to be *Meijer G-functions*, rather than the usual polynomial splines. Because Meijer G-functions are differentiable in their parameters, we can "tune" the parameters of the representation by gradient descent; as a consequence, our algorithm is efficient. Using five familiar data sets from the UCI repository and two familiar machine learning algorithms, we demonstrate that our algorithm produces *global* interpretations that are both *highly accurate* and *parsimonious* (involve a small number of terms). Our interpretations permit easy understanding of the relative importance of features and feature interactions. Our interpretation algorithm represents a leap forward from the previous state of the art.

## 1 Introduction

What do we need to trust the predictions of the black-box models crafted by our Machine Learning (ML) algorithms? Although the ML community has succeeded in generating accurate models in a very wide variety of settings, it has not yet succeeded in convincing most practitioners to adopt these models. One possible reason is that many practitioners will use familiar models over more accurate models that they do not understand. For a striking example, we cite the area of medical risk prediction, in which clinical models have remained the standard despite the fact that ML models are demonstrably more accurate. The explicitly stated reason for this is that an acceptable model must be both accurate and transparent [24]; state-of-the-art clinical models are transparent while state-of-the-art ML models are not. In responding to this challenge, one possibility would be to focus on the design of ML models that are themselves transparent; an alternative possibility is to make a given ML model more transparent by providing an *interpretation* of that model.

As detailed in Section 5, a substantial literature in the ML community follows the latter approach, but not entirely successfully. To understand the challenge that we are addressing here, it is instructive to make a simple thought experiment. Let us assume that the data follows a Cox proportional hazards model [10]. For such a model, the hazard rate at time $t$ is given by

$H(t \mid x_1, \ldots, x_d) = h_0(t) \exp\left(\sum_{i=1}^d b_i x_i\right)$, where $h_0(t)$ is a baseline hazard, which potentially depends on time, $x_1, \ldots, x_d$ are the features and the coefficients $b_1, \ldots, b_d \in \mathbb{R}$ represent the importance of each feature. Suppose that we are interested in identifying the part of this model which only depends on the features. This part is trivially given by $f(x) = \exp\left(\sum_{i=1}^d b_i x_i\right)$. We shall here consider this as the "black-box" that we would like to identify by using our interpretability method. If our interpretability method produces linear models, as it is the case for *LIME* [37], we could obtain an estimator $\hat{f}(x) = 1 + \sum_{i=1}^d b_i x_i$ for $f$. On the other hand, a polynomial estimator [2] for $f$ would be of the form $\hat{f}(x) = 1 + \sum_{i=1}^d b_i x_i/1! + (\sum_{i=1}^d b_i x_i)^2/2! + \textit{finite number of terms}$. It goes without saying that neither of those interpretable models captures an exponential relationship globally, hence precluding a global interpretation of the black-box model. Would it be possible to overcome these limitations of the state-of-the-art algorithms for interpretability? More precisely, is there a way to build a regressor allowing to capture a large class of interpretable functions *globally*? Needless to say that a positive answer to these questions would have a tremendous impact on the interpretability landscape in Machine Learning. To tackle this problem, we build a new approach on top of the two following corner stones.

**Projection Pursuit.** The projection pursuit algorithm is widely known in the statistics literature [13, 20, 22, 39]. Projection pursuit builds an approximation to a given function by proceeding in stages. In each stage, the algorithm finds a direction in the feature space and a ridge function of that direction that best approximate the residual between the given function and the approximation constructed in the previous state. The process continues until the desired degree of accuracy is achieved. In projection pursuit, the ridge functions are typically polynomial splines [18]. This reliance on polynomial splines means that prediction pursuit suffers from the very shortcoming noted above: it permits only local interpretations and hence does not suggest any global structure. For this reason, we should work with another family of functions.

**Meijer G-functions.** Instead of polynomial splines, we use functions from the class of Meijer G-functions. This class of functions includes all of the familiar functions used in modeling (polynomial, exponential, logarithmic, trigonometric and hypergeometric functions). A Meijer G-function is defined by four non-negative integer hyperparameters and an array of real parameters. An interesting property of these functions is that we can efficiently compute a numerical gradient of the Meijer G-functions with respect to these parameters. This allows to use gradient-based optimization [1].

**Contribution.** In this paper, we introduce a new algorithm called *Symbolic Pursuit* that produces an interpretable model for a given black-box. As in a projection pursuit, our algorithm gradually adds more terms in the model until the desired precision is achieved. In our analysis, we address a major difficulty introduced by using G-functions as ridge functions : the hyperparameters. As aforementioned, each G-function has four hyperparameters to tune. Fortunately, we show that there exist a set of five hyperparameter configurations that covers most familiar functions (polynomial, rational, exponential, trigonometric and hypergeometric functions). By restricting to these configurations, each G-function is optimized efficiently over a sufficiently large class of functions. Consequently, we demonstrate that our Symbolic Pursuit algorithm allows to produce highly accurate global models (that we call *symbolic models*) for black-boxes with a small number of G-functions. With our algorithm, one to five G-functions are enough to approximate a black-box model fitting a real world dataset. This is a leap forward from the previous state-of-the-art.

To make the paper self-contained, we start with a short presentation of the projection pursuit algorithm and Meijer G-functions in Section 2. We address the hyperparameter optimization problem of Meijer G-functions in Section 3. After these theoretical considerations, we assemble all the pieces to construct the Symbolic Pursuit algorithm in Section 4. We compare this algorithm to the state-of-the-art interpretability methods in Section 5. Finally, we demonstrate that our algorithm produces globally symbolic models with outstanding approximation power and parsimonious expressions on five real-world datasets in Section 6.

## 2 Mathematical preliminaries

We begin by recalling the projection pursuit algorithm, which we shall reshape for our purpose in Section 4. Then we review the definition of Meijer G-functions that we use as the building blocks of the Symbolic Pursuit algorithm.

## 2.1 Projection Pursuit

Let $\mathcal{X} = [0,1]^d$ be a normalized *feature space*, $\mathcal{Y} \subseteq \mathbb{R}$ be a *label space* and $f : \mathcal{X} \to \mathcal{Y}$ be a *black-box model* (e.g. a neural network). Because we view $f$ as a black-box, we do not know the true form of $f$, but we assume we can evaluate $f(x)$ for every $x \in \mathcal{X}$; i.e. we can *query* the black-box. The projection pursuit algorithm [13] constructs an *explicit* model $\hat{f}$ for $f$ of the form

$$\hat{f}(x) = \sum_{k=1}^{K} g_k\left(v_k^\top x\right), \tag{1}$$

where $v_k \in \mathbb{R}^d$ is a vector (onto which we project the feature vector $x \in \mathcal{X}$) and each $g_k$ is a function belonging to a specified class $\mathcal{F}$ of univariate functions. Because the functions $x \mapsto g_k\left(v_k^\top x\right)$ are constant on the hyperplanes normal to $v_k$, they are usually called *ridge functions*.

The approximation is built following an iterative process. At stage $j \in \mathbb{N}^*$, we suppose that the approximation $\hat{f}_j$ of the above form, but having only $j-1$ terms, has been constructed; i.e., $\hat{f}_j(x) = \sum_{k=1}^{j-1} g_k\left(v_k^\top x\right)$[1]. We define the residual $r_j = f - \hat{f}_j$ and find a vector $v_j$ and a function $g_j \in \mathcal{F}$ to minimize the $L^2$ norm of the updated residual:

$$(g_j, v_j) = \arg\min_{\mathcal{F} \times \mathbb{R}^d} \int_{\mathcal{X}} \left[r_j(x) - g\left(v^\top x\right)\right]^2 dF(x) \tag{2}$$

where $F$ is the cumulative distribution function on the feature space. This process is continued until the $L^2$ norm of the residual is smaller than a predetermined threshold. This algorithm is typically complemented with a back-fitting strategy in practice [17].

In principle, projection pursuit would seem an excellent approach to interpretability because each stage provides one term in the approximation and the process can be terminated at any stage. Hence projection pursuit allows a trade-off between the accuracy of the representation $\hat{f}$ and the complexity of $\hat{f}$. In practice, however, the class $\mathcal{F}$ is usually taken to be the class of polynomial splines – often of low degree (e.g. cubic) [20]. This has the advantage of making each stage of the algorithm computationally tractable, but the disadvantage of often leading to representations that involve many terms. Moreover, often even the individual terms in these representations do not have natural interpretations. These considerations suggest looking for a different class of functions that retain computational tractability while leading to representations that are more parsimonious and interpretable.

## 2.2 Meijer G-functions

Is there a set of functions $\mathcal{F}$ that includes most interpretable functions and that would allow solving the optimization problem (2) with standard techniques? The answer to this question is yes. The set of Meijer G-function, that we denote $\mathcal{G}$, fulfils these two requirements, as discussed in [1]. We shall thus henceforth restrict our investigations to $\mathcal{F} = \mathcal{G}$. Here, we briefly recall the definition of a Meijer G-function [5].

**Definition 2.1** (Meijer G-function [5]). A Meijer G-function is defined by an integral along a path $\mathcal{L}$ in the complex plane,

$$G_{p,q}^{m,n}\left(\begin{array}{c} a_1, \ldots, a_p \\ b_1, \ldots, b_q \end{array} \middle| z\right) = \frac{1}{2\pi i} \int_{\mathcal{L}} z^s \frac{\prod_{j=1}^{m} \Gamma(b_j - s) \prod_{j=1}^{n} \Gamma(1 - a_j + s)}{\prod_{j=m+1}^{q} \Gamma(1 - b_j + s) \prod_{j=n+1}^{p} \Gamma(a_j - s)} ds, \tag{3}$$

where $a_i, b_j \in \mathbb{R}$ ; $\forall i = 1, \ldots, p$ ; $j = 1, \ldots, q$ ; $m, n, p, q \in \mathbb{N}$ with $m \leq q$ , $n \leq p$ and $\Gamma$ is Euler's Gamma function. The path of integration $\mathcal{L}$ is chosen so that the poles associated to the two families of Gamma functions (one family for the $a$'s and one family for the $b$'s) lie on different sides of $\mathcal{L}$.[2] The definition yields a complex-analytic function in the entire complex plane $\mathbb{C}$ with the possible exception of the origin $z = 0$ and the unit circle $\{z \in \mathbb{C} : |z| = 1\}$. We will only consider its behavior for *real $z$* in the open unit interval $(0,1) \subset \mathbb{R}$. We write $\mathcal{G}$ for the class of all Meijer G-functions.

This set of function is interesting because it includes most familiar functions such as exponential, trigonometric and hypergeometric functions as particular cases [31]. For example, the negative exponential function is

$$\exp(-x) = G_{0,1}^{1,0} \left( \begin{array}{c} - \\ 0 \end{array} \middle| \ x \ \right).$$

Furthermore, we can compute a numerical gradient of these function with respect to the real parameters $a_i, b_j$, hence allowing the use of gradient-based optimization. In the following, it will be useful to denote $\mathcal{G}_{p,q}^{m,n}$ the set of Meijer G-functions of hyperparameters $m, n, p, q$. The full set of Meijer G-function can thus be decomposed as

$$\mathcal{G} = \bigcup_{p,q \in \mathbb{N}} \bigcup_{n=0}^{p} \bigcup_{m=0}^{q} \mathcal{G}_{p,q}^{m,n}. \tag{4}$$

It should already be obvious at this stage that we won't be able to search in all of these subsets of $\mathcal{G}$ so that $m, n, p, q$ will have to be fixed as hyperparameters. How much are we restricting the possibilities by doing so? Is there a clever restriction that would include most familiar functions? We will elaborate on these questions in Section 3.

## 3 Hyperparameters climbing down the trees

In this section, we address a major theoretical challenge related to the use of Meijer G-functions: the number of hyperparameters. More precisely, we have 4 hyperparameters $(m, n, p, q) \in \mathbb{N}^4$ to tune for each Meijer G-function appearing in the expansion (1). It should be obvious that we cannot search across every subclass $\mathcal{G}_{p,q}^{m,n}$. Fortunately, this is not necessary; we can choose a set $\mathbb{H}$ of 4-tuples of hyperparameters that is large enough that the subclasses $\mathcal{G}_{p,q}^{m,n}; (m, n, p, q) \in \mathbb{H}$ encompass sufficiently many functions but small enough that searching within and across these subclasses is computationally tractable. Our choice of $\mathbb{H}$ relies on two following results.

First, we note that the subsets $\mathcal{G}_{p,q}^{m,n}$ of $\mathcal{G}$ associated with different values of the hyperparameters are not disjointed. Indeed, we show that the following result holds:

**Lemma 3.1.** *For all* $(m, n, p, q) \in \mathbb{N}^4$ *such that* $p, q \geq 1$:

- *If* $m \geq 1$: $\mathcal{G}_{p-1,q-1}^{m-1,n} \subset \mathcal{G}_{p,q}^{m,n}$

- *If* $n \geq 1$: $\mathcal{G}_{p-1,q-1}^{m,n-1} \subset \mathcal{G}_{p,q}^{m,n}$

*Proof.* A detailed proof can be found in Section 1 of the supplementary material. ∎

This important lemma tells that, during the optimization, we can explore different values of $(m, n, p, q)$ than the one initially fixed. To illustrate, consider the 4-tuple $(m, n, p, q) = (1, 1, 2, 2)$. Lemma 3.1 implies that $\mathcal{G}_{2,2}^{1,1}$ contains both $\mathcal{G}_{1,1}^{0,1}$ and $\mathcal{G}_{1,1}^{1,0}$, and that each of these in turn contains $\mathcal{G}_{0,0}^{0,0}$, as in Figure 1. Therefore, starting with $(m, n, p, q) = (1, 1, 2, 2)$ also allows to explore $(m, n, p, q) = (0, 1, 1, 1), (1, 0, 1, 1), (0, 0, 0, 0)$ at the same time. This concept can be generalized so that the subsets of $\mathcal{G}$ can be represented in terms of trees of inclusion, such as the one from Figure 1. Building on this idea

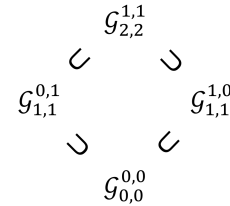

Figure 1: The tree of inclusions starting from $\mathcal{G}_{2,2}^{1,1}$.

of trees of inclusion, we propose to choose a clever finite set of configurations for the hyperparameters which allows to recover most closed form expressions.

**Proposition 3.1.** *Consider the set of Meijer G-functions of the form*

$$\hat{f}(z) = G_{p,q}^{m,n} \left( \begin{array}{c} a_1, \ldots, a_p \\ b_1, \ldots, b_q \end{array} \middle| \ \begin{array}{c} s \cdot \\ z^r \end{array} \right), \tag{5}$$

*where $a_1, \ldots, a_p, b_1, \ldots, b_q \in \mathbb{R}$ ; $r, s \in \mathbb{R}$ and the hyperparameters belong to the configuration set $(m, n, p, q) \in \mathbb{H} = \{(1, 0, 0, 2), (0, 1, 3, 1), (2, 1, 2, 3), (2, 2, 3, 3), (2, 0, 1, 3)\}$. This set of function includes all the functions with the form*

$$f(z) = \Phi(w \cdot z^q) \cdot z^t, \tag{6}$$

*with $w, q, t \in \mathbb{R}$ ; $\Phi \in \left\{ \mathrm{id}, \sin, \cos, \sinh, \cosh, \exp, \log(1 + \cdot), \arcsin, \arctan, J_\nu, Y_\nu, I_\nu, \frac{1}{1+\cdot}, \Gamma \right\}$ where $J_\nu, Y_\nu, I_\nu$ are the Bessel functions and $\Gamma$ is Euler's Gamma function.*

*Remark.* The above set of Meijer G-function includes much more function than the one depicted in (6). The purpose of the above proposition is only to suggest the generality of the symbolic model that we can assemble by using these Meijer G-functions as building blocks. Here, we note that our choice allows to cover many exponential, trigonometric, rational and Bessel functions.

*Proof.* A detailed proof can be found in Section 1 of the supplementary material. □

This theorem gives a very satisfying prescription for the hyperparameters. Building on this, we now have a realistic restriction of $\mathcal{G}$ that we can use for the set $\mathcal{F}$ appearing in (2). We shall henceforth use the following notation for this restriction:

$$\mathcal{G}_{\mathbb{H}} = \bigcup_{(m,n,p,q) \in \mathbb{H}} \mathcal{G}_{p,q}^{m,n}. \tag{7}$$

Therefore, our Symbolic Pursuit algorithm will search over the set of functions $\mathcal{F} = \mathcal{G}_{\mathbb{H}}$. All the ingredients are now ready to build the algorithm, this is the subject of next section.

## 4   Symbolic Pursuit

In this section, we build our *Symbolic Pursuit* algorithm. With all the ingredients we have prepared in the previous sections, this will be straightforward. The starting point is naturally to consider the functions $g_1, \ldots, g_K$ appearing in (1) to be elements of $\mathcal{G}_{\mathbb{H}}$. However, we have to be careful because Meijer G-functions might not be defined when $z = 0, 1$. By looking at (1), we note that the arguments of each function $g_k$ takes the form $v_k^\top x$ for $k = 1, \ldots, K$. We shall now scale this linear combination so that it stays in the domain $(0, 1)$ of the Meijer-G function $g_k$.

The Cauchy-Schwartz inequality for the $l^2$ inner product guarantees that $|v_k^\top x| \leq \|v_k\|.\|x\|$. Our normalization for $x \in \mathcal{X} = [0, 1]^d$ guarantees that $\|x\| \leq \sqrt{d}$. By mixing these two ingredients, we get

$$\frac{|v_k^\top x|}{\|v_k\|\sqrt{d}} \leq 1. \tag{8}$$

Therefore, we can simply take the *ReLU* of $\frac{v_k^\top x}{\|v_k\|\sqrt{d}}$ as an admissible argument for the G-function $g_k$[3]. In conclusion, we make the following replacement in (1):

$$v_k^\top x \to \left( \frac{v_k^\top x}{\|v_k\|\sqrt{d}} \right)^+ \equiv \max \left( 0, \frac{v_k^\top x}{\|v_k\|\sqrt{d}} \right). \tag{9}$$

Note that, in this way, the argument remains a linear combination of the features in the region where this linear combination is positive[4].

Because our optimization problem is non-convex, it is helpful to allow our pursuit algorithm to correct the output of previous iterations at each iteration via a back-fitting strategy. A conventional way to implement this is to add a weight $w_k$ in front of each term in the expansion (1) and optimize this weight during the back-fitting procedure [15]. This will guarantee that terms that are constructed

at some iteration of the algorithm and found to be largely irrelevant at some later iteration will be assigned a small weight. We can now rewrite the symbolic model (1) as

$$\hat{f}(x) = \sum_{k=1}^{K} w_k \cdot g_k \left( \left[ \frac{v_k^\top x}{\|v_k\|\sqrt{d}} \right]^+ \right).$$ (10)

Similarly, we can reformulate the optimization problem (2) as

$$(g_k, v_k, w_k) = \underset{\mathcal{G}_\mathbb{H} \times \mathbb{R}^d \times \mathbb{R}}{\arg\min} \int_{\mathcal{X}} \left[ r_k(x) - w \cdot g \left( \left[ \frac{v^\top x}{\|v\|\sqrt{d}} \right]^+ \right) \right]^2 dF(x).$$ (11)

Note that the optimization over $g_k$ takes into account the five hyperparameters configuration. The objective function in (11) is solved by trying each configuration and keeping the one associated to the smallest loss. At each step of the projection pursuit algorithm, we shall use a back-fitting strategy to correct all the terms that already appear in the expansion. Keeping this in mind, the back-fitting for the term $l \in \{1, \dots, k-1\}$ will consist in minimizing the residue which excludes the contribution of term $l$ at iteration $k$

$$r_{k,l}(x) \equiv f(x) - \sum_{j \neq l}^{k} w_j \cdot g_j \left( \left[ \frac{v_j^\top x}{\|v_j\|\sqrt{d}} \right]^+ \right).$$ (12)

The pseudo code of Symbolic Pursuit is provided in Section 5 of the supplementary material, in which we write our algorithm that solves this optimization problem step by step. It should be stressed that, on a mathematical ground, using Meijer G-functions in optimization problems is far from anodyne. We elaborate on some theoretical impact of this choice on the behavior of the loss function in Section 3 of the supplementary material.

## 5   Related work

Table 1: Symbolic Pursuit and the state-of-the-art interpretability methods.

| Algorithm | Feature Importance | Feature Interaction | Model Independent | Global | Parsimonious |
|---|:---:|:---:|:---:|:---:|:---:|
| GA$^2$M [30] | ✓ | ✓ | ✓ | | |
| LIME [37] | ✓ | | ✓ | | |
| SHAP [32] | ✓ | ✓ | ✓ | | |
| DeepLIFT [40] | ✓ | | | | |
| L2X [7] | ✓ | | ✓ | | |
| NIT [42] | ✓ | ✓ | | | |
| INVASE [45] | ✓ | | ✓ | | |
| Symb. Metamodel [1] | ✓ | ✓ | ✓ | ✓ | |
| **Symbolic Pursuit** | ✓ | ✓ | ✓ | ✓ | ✓ |

In this section, we compare our algorithm to some of the most popular state-of-the-art interpretability methods, this discussion is summarized in Table 1. Like most methods, our Symbolic Pursuit algorithm allows to learn about feature importance and feature interaction for each prediction. We shall give more details about these two points in Section 6. Unlike methods such as *DeepLIFT* [40] or *NIT* [42], our method is model independent (i.e. not specialized for a particular class of black-box), as we shall demonstrate in Section 6. However, unlike the other methods, which provide only local information, Symbolic Pursuit also provides global information. This comes from the fact that Meijer-G functions allow to capture a large class of closed form expressions globally, as we detailed in Section 1 and 3. The only other interpretive method that provides global information is the method of Symbolic Metamodeling [1], which also makes use of Meijer G-functions. However, this last method fails to produce parsimonious expressions since it produces an additive model

$$\hat{f}(x_1, \dots, x_d) = \sum_{i=1}^{d} g_i(x_i) + \sum_{i=1}^{d} \sum_{j<i} g_{ij}(x_i \cdot x_j),$$ (13)

where the $g$'s are Meijer G-function. This model is made of $d + {}^{d(d-1)}/_2 = {}^{d(d+1)}/_2$ terms. For ten features $d = 10$, this corresponds to 55 terms. The size of the Symbolic metamodel enormously increases the burden of constructing interpretations, which makes it hard to use in most practical situations. Moreover, only one hyperparameter configuration is explored for all of these terms since all the terms are optimized simultaneously. In the language that we have introduced in Section 3, this means that only one hyperparameter tree can be explored for all the G-functions $g$'s. Indeed, exploring all the hyperparameter configurations that we have identified in Section 3 would require to solve $|\mathbb{H}|^{d(d+1)/2} = 5^{d(d+1)/2}$ optimization problems corresponding to each choice of hyperparameter configuration for each $g$. This is unrealistic when $d$ is large. Therefore, it is not possible to identify all the familiar functions simultaneously, since these are associated to different hyperparameter trees[5].

To solve these issues, we introduced the *Symbolic Pursuit* algorithm which benefits from the ability to produce parsimonious expressions. More precisely, this method encapsulates perfectly the trade-off between accuracy and interpretability as larger (and therefore less interpretable) expressions will typically be more accurate. Therefore, parsimony naturally translates into stopping the optimization when a reasonable accuracy has been achieved. We will show in Section 6 that, even for real world datasets, this does not require a large number of terms.

## 6 Experiments

### 6.1 Synthetic data

In this section, we evaluate the performance of Symbolic Pursuit on several synthetic datasets [6]. In order to compare the performance of our method against popular interpretability methods, we shall restrict to explanations in the form of feature importance. More precisely, we focus on a pseudo black-box $f$ for which the feature importance is known unambiguously. Here, we consider a three dimensional linear model

$$f(x_1, x_2, x_3) = \beta_1 \cdot x_1 + \beta_2 \cdot x_2 + \beta_3 \cdot x_3 = \beta^\top x,$$

where $\beta = (\beta_1, \beta_2, \beta_3) \in \mathbb{R}^3$ and $x = (x_1, x_2, x_3) \in [0, 1]^3$. Since this pseudo black-box is merely a linear model, we expect the interpretability methods to output an importance vector of $\beta$ for each test point. Because only relative importance matters in practice, we assume that all the importance vectors are normalized so that $\beta \in S^2$ where $S^2$ is the 2-sphere.

In our experiment, we start by drawing a true importance vector $\beta \sim U([1, 10]^3)$ that we normalize subsequently. Then, we build a Symbolic model, a *LIME* explainer [37] and a *SHAP* explainer [32] for $f$. Finally, we draw 30 test points $x_{test} \sim U([0, 1]^3)$ that we input to the three interpretability methods to build an estimator $\hat{\beta}$ for $\beta$. We reproduce this experiment 100 times and report the MSE between $\hat{\beta}$ and $\beta$ for each interpretability method in Table 2. We note

| Method | $\|\beta - \hat{\beta}\|^2$ |
|---|---|
| LIME | $0.66 \pm 0.07$ |
| SHAP | $0.66 \pm 0.07$ |
| **Symbolic** | $0.02 \pm 0.05$ |

Table 2: Symbolic Pursuit on synthetic data.

similar performances of SHAP and LIME and a significant improvement with Symbolic Models.

### 6.2 Real world data

In this section, we evaluate the performance of Symbolic Pursuit on two popular black-box models – a Multilayer Perceptron (MLP) and Support Vector Machine (SVM) – applied to five UCI datasets [12] including Wine Quality Red (Wine) and Yacht Hydrodynamics (Yacht), Boston Housing (Boston), Energy Efficiency (Energy) and Concrete Strength (Concrete). Both models are implemented using the `scikit-learn` library [6] with the default hyperparameters. For each experiment, we split the dataset into a training set ($80\%$) and a test set ($20\%$). The training set is used to produce the Black-Box model and the Symbolic Model (via a mixup strategy for the later, as detailed in Section

5 of the supplementary material); the test set is used to test the model performance. We repeat the experiment five times with different random splits and report averages and standard deviations. Table 3 shows the mean squared error (MSE) and $R^2$ of the black-box models against the true labels, the approximation MSE and $R^2$ of the symbolic models against the corresponding black-box models, the MSE and $R^2$ of the symbolic model against the true labels, and the number of terms in the symbolic models constructed for each of the five random splits.[7] The precise definition of the metrics we use here can be found in Section 6 of the supplementary material. As can be seen in Table 3 (Symbolic vs Black-box), all the symbolic models achieve low MSE and high $R^2$ with respect to the black-box model. Indeed, in nine of ten cases the $R^2$ is above 0.95, and in the tenth it is above 0.90. Moreover, in 15/50 instances, the symbolic model has only a single term, in 36/50 instances the symbolic model has at most two terms and in 48/50 instances the symbolic model has at most three terms. This provides strong evidence that Symbolic Pursuit produces interpretations that are both accurate and parsimonious, as previously asserted.

Table 3: Symbolic Pursuit results on the five UCI datasets [12].

| Models | Datasets | Black-box | | Symbolic vs Black-box | | Symbolic | | # Terms |
|---|---|---|---|---|---|---|---|---|
| | | MSE | $R^2$ | MSE | $R^2$ | MSE | $R^2$ | |
| MLP | Wine | $0.016 \pm .002$ | $0.179 \pm .061$ | $0.001 \pm .001$ | $0.964 \pm .018$ | $0.016 \pm .002$ | $0.179 \pm .054$ | 1, 1, 1, 2, 1 |
| | Yacht | $1.433 \pm .681$ | $0.426 \pm .233$ | $0.008 \pm .014$ | $0.978 \pm .023$ | $1.458 \pm .653$ | $0.413 \pm .225$ | 2, 2, 2, 1, 2 |
| | Boston | $0.050 \pm .015$ | $0.681 \pm .063$ | $0.004 \pm .001$ | $0.952 \pm .021$ | $0.053 \pm .016$ | $0.660 \pm .066$ | 1, 3, 3, 3, 2 |
| | Energy | $0.015 \pm .001$ | $0.926 \pm .012$ | $0.002 \pm .001$ | $0.988 \pm .004$ | $0.016 \pm .003$ | $0.918 \pm .013$ | 2, 2, 2, 1, 2 |
| | Concrete | $0.100 \pm .006$ | $0.538 \pm .061$ | $0.001 \pm .000$ | $0.988 \pm .003$ | $0.100 \pm .005$ | $0.533 \pm .056$ | 1, 2, 3, 2, 3 |
| SVM | Wine | $0.014 \pm .001$ | $0.331 \pm .026$ | $0.001 \pm .001$ | $0.904 \pm .038$ | $0.014 \pm .001$ | $0.301 \pm .039$ | 1, 2, 1, 1, 1 |
| | Yacht | $0.723 \pm .179$ | $0.555 \pm .036$ | $0.010 \pm .015$ | $0.973 \pm .043$ | $0.737 \pm .197$ | $0.547 \pm .050$ | 3, 1, 2, 2, 2 |
| | Boston | $0.040 \pm .005$ | $0.740 \pm .053$ | $0.001 \pm .001$ | $0.984 \pm .017$ | $0.043 \pm .006$ | $0.724 \pm .046$ | 3, 3, 2, 1, 2 |
| | Energy | $0.015 \pm .002$ | $0.928 \pm .007$ | $0.002 \pm .003$ | $0.985 \pm .016$ | $0.018 \pm .002$ | $0.913 \pm .006$ | 2, 3, 2, 3, 2 |
| | Concrete | $0.069 \pm .005$ | $0.676 \pm .039$ | $0.003 \pm .002$ | $0.971 \pm .015$ | $0.082 \pm .011$ | $0.623 \pm .043$ | 3, 5, 3, 1, 5 |

In this setup, it is possible to do algebraic manipulations on a symbolic model to extract transparent information of the black-box model. For instance, we could simply inspect the components of each vector $v_k$ to get an idea of the feature importance. We could also differentiate the G-function $g_k$ with respect to the features to obtain the gradients in closed form expression. Most importantly, it is realistic to do these operations by hand at this stage since the expressions are short, the Meijer G-functions can be differentiated with respect to their argument easily [31] and their arguments are linear combinations of features. To illustrate, we use one of the interpretations of MLP on the Wine dataset; in order not to make the task too easy, we use the split that produces two terms in the interpretation, rather than any of the splits that produce a single term. With our notations, $f$ denotes the MLP black-box and $\hat{f}$ denotes its symbolic model. Because $\hat{f}$ has two terms, it can be written as

$$\hat{f}(x) = w_1 g_1 \left( \frac{v_1^\top x}{\|v_2\| \sqrt{d}} \right) + w_2 g_2 \left( \frac{v_2^\top x}{\|v_2\| \sqrt{d}} \right). \tag{14}$$

By construction, $g_1, g_2 \in \mathcal{G}_\mathbb{H}$ but in this instance $g_1, g_2$ do not appear to have expressions in terms of familiar functions. Despite this, we can easily extract useful information from (14) by building local polynomial models via a Taylor expansion of $g_1$ and $g_2$, respectively. Let $z_j = \frac{v_j^\top x}{\|v_j\| \sqrt{d}}$ for $j = 1, 2$. Using the Taylor expansion of $g_1$ and $g_2$, we produce the first order Taylor expansion of the symbolic model $\hat{f}(x)$ around an instance $x$ from the test set:

$$\hat{f}_1(x) = \tilde{v}^\top x + \tilde{c} = c_0 + c_{1,1} z_1 + c_{1,2} z_2. \tag{15}$$

In this particular instance $x$, we find that $\tilde{c} = 0.8399$, $c_0 = 1.001$, $c_{1,1} = -0.2339$, $c_{1,2} = 0.3280$; the values of $\tilde{v}, v_1, v_2$ are displayed in Figure 2 (b-d).

Let us now compare this linear model offered by $\hat{f}_1(x)$ with a local linear surrogate model computed with *LIME* [37]. LIME explainer suggests that the most important features for the MLP model at $x$ are $x_{10}$ (alcohol), $x_9$ (sulphates) and $x_8$ (pH) (See Figure 1 of the supplementary material). The first order interpreter $\hat{f}_1(x)$ in (15) agrees with this, since $x_8$, $x_9$ and $x_{10}$ have the highest weight

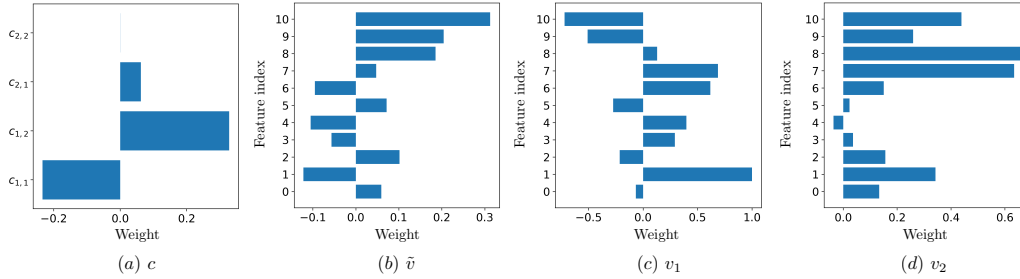

Figure 2: Coefficients and feature weights in our symbolic model.

in $\tilde{v}$, as shown in Figure 2 (b). However, the agreement is not perfect since LIME also suggests that $x_5$ (free sulfur dioxide) and $x_6$ (total sulfur dioxide) are important and our interpreter does not. Unlike LIME, our interpreter can easily provide a second-order Talor expansion of $g_1$ and $g_2$ to suggest important interactions *between features*. The second-order Taylor expansion of $\hat{f}(x)$ has the form $\hat{f}_2(x) = \hat{f}_1(x) + c_{2,1}z_1^2 + c_{2,2}z_2^2 \approx \hat{f}_1(x) + c_{2,1}(v_1^\top x)^2$. In this case we find that $\tilde{c} = 0.8399$ $c_{2,1} = 0.0623$ and $c_{2,2} = -0.0002$ so that the interactions appearing in $z_1^2$ are important compared to the interactions in $z_2^2$. Hence we can see from Figure 2 (c) that $x_1$ (fixed acidity) has important interactions with features $x_7$ (density), $x_9$ and $x_{10}$ but not $x_8$, despite the fact that $x_8$ itself is an important feature. This short discussion allows to see that the major advantage of our method compared to LIME is that we can go beyond a linear model to capture nonlinear properties of the black-box, such as feature interactions.

## 7 Conclusion

This paper has proposed an algorithm that produces a *global* interpretation of any given continuous black-box function. Our algorithm employs a variation of projection pursuit in which the ridge functions are chosen to be *Meijer G-functions*, rather than the usual polynomial splines. A series of experiments demonstrates that the interpretations produced by our algorithm are both accurate and parsimonious, and that the interpretation yields more information than is available from other methods. Because our method produces continuous models, it may not be appropriate for the interpretation of discontinuous black-box models such as tree-based models, at least without some adaptation and/or qualification. Perhaps more importantly, although our interpretive models are parsimonious, they frequently involve unfamiliar functions. We have argued above that this is not necessarily a barrier to understanding, because the explicit expressions can be easily used to produce local interpretations in terms of linear functions or low-order polynomials that reveal which features and which interactions between features are most important. In the medical domain, for example, this information is often very valuable, but not easily obtained.

## Broader Impact

As mentioned at the very beginning of this paper, the lack of easy interpretation of ML models has proved a serious obstacle to their adoption – despite their demonstrated accuracy. Because ML models are more accurate than previous models, anything that makes ML models more widely used is likely to have enormous positive affects in practice – simply by providing better predictions. Even interpretations that provide only first-order information will surely prove to be important – not least by allowing for easier correction of measurement/recording errors. Two examples may illustrate. (1) It is well-documented that "No-Fly" lists often flag the wrong people who happen to have the same names as the right people, and that getting removed from such lists can be a difficult task. (2) It is similarly well-documented that measurement/recording errors are common in the calculation of credit scores – but because those calculations are often quite opaque, such errors are often left uncorrected, so some who should be approved for loans are declined, and others who should be declined are approved .

This paper has offered a method for producing accurate and parsimonious interpretations of black-box models. In no sense do we view this as providing the final or best method for interpretation; indeed we view this work as taking, along with [1], only the first few steps in a new and very promising direction. We have already noted that, because the interpretations our algorithm produces are continuous, our method may not be suitable for interpretation of black-box models such as Decision Trees or Random Forests, and may not be suitable for classification problems unless they are transformed into regression problems by assigning probabilities instead of decisions. In itself, this transformation is simple and unobjectionable, but clients who expect "Buy" or "Sell" advice from their financial advisor may not be happy with a recommendation to "Buy with probability 0.7 and Sell with probability 0.3."

Despite the fact that our symbolic models contain few terms, which is a significant improvement compared to [1], we still have to deal with some Meijer G-functions or hypergeometric functions that don't reduce to familiar expressions. If this family of functions is explicitly used in some scientific communities, such as in physics [14], they are not likely to be deemed interpretable by some practitioners. A possible way to deal with this issue would be to enforce the cancellation between some zeroes and poles of the Meijer G-functions. This can be done by adding a lasso penalty to the loss, this approach is detailed in Section 4 of the supplementary material. We are convinced that the Symbolic Pursuit algorithm opens up several interesting research paths in the ML intepretability landscape. We hope that this paper will convince the ML community to walk along them to explore this emerging paradigm of interpretability.

## Acknowledgments

Jonathan Crabbe is supported by Aviva. Yao Zhang is supported by GSK. Mihaela van der Schaar is supported by the Office of Naval Research (ONR), NSF 1722516.

## Footnotes

[1] If $j = 1$ then $\hat{f}_1 \equiv 0$.

[2] We omit some technical details and restrictions; see [5] for details.

[3]This is not formally true since the inequality in (8) is not strict. However, this is not important in practice since we can perfectly normalize the features to a closed interval included in (0,1).

[4]Which would not be the case if we had used a sigmoid to restrict the range of the argument.

[5]For instance $x \mapsto \exp(-x) \in \mathcal{G}_{0,1}^{1,0}$ and $x \mapsto \ln(1+x) \in \mathcal{G}_{2,2}^{1,2}$ and these two sets cannot be put in a same tree of inclusion via Lemma 3.1.

[6]The code for Symbolic Pursuit is available at https://bitbucket.org/mvdschaar/mlforhealthlabpub and https://github.com/JonathanCrabbe/Symbolic-Pursuit.

[7] If the symbolic model is a good approximation of the black-box model, it will necessarily have a similar MSE and $R^2$ against the true labels, as shown in the Black-box and Symbolic column of Table 3.

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
