[Supplementary Material]

# Supplementary Material for Learning outside the Black-Box: The pursuit of interpretable models

**Jonathan Crabbe**
University of Cambridge
jc2133@cam.ac.uk

**Yao Zhang**
University of Cambridge
yz555@cam.ac.uk

**William R. Zame**
University of California Los Angeles
zame@econ.ucla.edu

**Mihaela van der Schaar**
University of Cambridge
University of California Los Angeles
The Alan Turing Institute
mv472@cam.ac.uk

## 1 Hyperparameters climbing down the tree, the proofs

In this section, we prove the propositions appearing in the paper.

### 1.1 The lemmas

**Lemma 1.1.** *For all natural numbers $m, n, p, q$ such that $p, q \geq 1$:*

- *If $m \geq 1 : \mathcal{G}_{p-1,q-1}^{m-1,n} \subset \mathcal{G}_{p,q}^{m,n}$*

- *If $n \geq 1 : \mathcal{G}_{p-1,q-1}^{m,n-1} \subset \mathcal{G}_{p,q}^{m,n}$*

*Proof.* This lemma is a trivial consequence of the definition of Meijer G-functions. Let us give a proof the first proposition. For all $f \in \mathcal{G}_{p-1,q-1}^{m-1,n}$, there exist reals $a_1, \ldots, a_{p-1}, b_1, \ldots, b_{q-1}$ such that $\forall x \in (0, 1)$:

$$
\begin{aligned}
f(x) &= G_{p-1,q-1}^{m-1,n} \left( \begin{array}{c} a_1, \ldots, a_{p-1} \\ b_1, \ldots, b_{q-1} \end{array} \middle| x \right) \\
&= \frac{1}{2\pi i} \int_{\mathcal{L}} ds \, x^s \frac{\prod_{j=1}^{m-1} \Gamma(b_j - s) \prod_{j=1}^{n} \Gamma(1 - a_j + s)}{\prod_{j=m+1}^{q-1} \Gamma(1 - b_j + s) \prod_{j=n+1}^{p-1} \Gamma(a_j - s)}
\end{aligned}
$$

We shall show that this G-function can be represented in terms of another G-function $\tilde{f} \in \mathcal{G}_{p,q}^{m,n}$. Let $c \in \mathbb{R} \setminus \{a_1, \ldots, a_{p-1}, b_1, \ldots b_{q-1}\}$, consider the following choice for $\tilde{f}$:

$$
\begin{aligned}
\tilde{f}(x) &= G_{p,q}^{m,n} \left( \begin{array}{c} a_1, \ldots, a_{p-1}, c \\ c, b_1, \ldots, b_{q-1} \end{array} \middle| x \right) \\
&= \frac{1}{2\pi i} \int_{\tilde{\mathcal{L}}} ds \, x^s \frac{\Gamma(c - s) \times \prod_{j=1}^{m-1} \Gamma(b_j - s) \prod_{j=1}^{n} \Gamma(1 - a_j + s)}{\prod_{j=m+1}^{q-1} \Gamma(1 - b_j + s) \prod_{j=n+1}^{p-1} \Gamma(a_j - s) \times \Gamma(c - s)} \\
&= G_{p-1,q-1}^{m-1,n} \left( \begin{array}{c} a_1, \ldots, a_{p-1} \\ b_1, \ldots, b_{q-1} \end{array} \middle| x \right) = f(x).
\end{aligned}
$$

The only nontrivial step in the above reasoning is going from the second to the third line. Indeed, we have to show that the paths $\mathcal{L}$ and $\tilde{\mathcal{L}}$ are compatible with each other. In both cases, the contour

appearing in the definition of the Meijer G-function should separate the sequence of poles[1] going to $-\infty$ from those going to $+\infty$ [3]. We note that, because of our choice for $c$, the poles associated to $\Gamma(c - s)$ in the integral defining $\tilde{f}$ are not poles of the integral defining $f$. Therefore, it follows from the residue theorem [1] that we can reshape $\mathcal{L}$ into $\tilde{\mathcal{L}}$ without affecting the result of the integral defining $f$. This proves that $f = \tilde{f} \in \mathcal{G}_{p,q}^{m,n}$ for any choice of the above parameter $c$. The proof of the second proposition follows by interchanging the roles of the $a$'s and the $b$'s. $\qquad\square$

**Lemma 1.2.** *For all $(m, n, p, q) \in \mathbb{N}^4$ and $t \in \mathbb{R}$, the set $\mathcal{G}_{p,q}^{m,n}$ has the following closure property: If $f \in \mathcal{G}_{p,q}^{m,n}$ then the function $x \mapsto f(x).x^t \in \mathcal{G}_{p,q}^{m,n}$.*

*Proof.* For all $f \in \mathcal{G}_{p,q}^{m,n}$, there exist reals $a_1, \ldots, a_p, b_1, \ldots, b_q$ such that $\forall x \in (0, 1)$:

$$
\begin{aligned}
f(x) &= G_{p,q}^{m,n} \left( \begin{array}{c} a_1, \ldots, a_p \\ b_1, \ldots, b_q \end{array} \middle| \ x \right) \\
&= \frac{1}{2\pi i} \int_{\mathcal{L}} ds \ x^s \frac{\prod_{j=1}^{m} \Gamma(b_j - s) \prod_{j=1}^{n} \Gamma(1 - a_j + s)}{\prod_{j=m+1}^{q} \Gamma(1 - b_j + s) \prod_{j=n+1}^{p} \Gamma(a_j - s)}.
\end{aligned}
$$

We note that, for all $t \in \mathbb{R}$, we have

$$
f(x).x^t = \frac{1}{2\pi i} \int_{\mathcal{L}} ds \ x^{s+t} \frac{\prod_{j=1}^{m} \Gamma(b_j - s) \prod_{j=1}^{n} \Gamma(1 - a_j + s)}{\prod_{j=m+1}^{q} \Gamma(1 - b_j + s) \prod_{j=n+1}^{p} \Gamma(a_j - s)}.
$$

Let us introduce the new integration variable $\tilde{s} = s + t$. Trivially, we have that $ds = d\tilde{s}$ so that the integral can be rewritten

$$
f(x).x^t = \frac{1}{2\pi i} \int_{\tilde{\mathcal{L}}} d\tilde{s} \ x^{\tilde{s}} \frac{\prod_{j=1}^{m} \Gamma(b_j + t - \tilde{s}) \prod_{j=1}^{n} \Gamma(1 - a_j - t + \tilde{s})}{\prod_{j=m+1}^{q} \Gamma(1 - b_j - t + \tilde{s}) \prod_{j=n+1}^{p} \Gamma(a_j + t - \tilde{s})},
$$

where $\tilde{\mathcal{L}}$ is the image of $\mathcal{L}$ under the translation $z \mapsto z + t$ in the complex plane. It follows form this, and the definition of Meijer-G functions that

$$
f(x).x^t = G_{p,q}^{m,n} \left( \begin{array}{c} a_1 + t, \ldots, a_p + t \\ b_1 + t, \ldots, b_q + t \end{array} \middle| \ x \right).
$$

We conclude that the function $x \mapsto f(x).x^t$ is an element of $\mathcal{G}_{p,q}^{m,n}$, which achieves the proof. $\qquad\square$

## 1.2 The proposition

**Proposition 1.1.** *Consider the set of Meijer G-functions of the form*

$$
\hat{f}(z) = G_{p,q}^{m,n} \left( \begin{array}{c} a_1, \ldots, a_p \\ b_1, \ldots, b_q \end{array} \middle| \ s.z^r \right), \tag{1}
$$

*where $a_1, \ldots, a_p, b_1, \ldots, b_q \in \mathbb{R}$ ; $r, s \in \mathbb{R}$ and the hyperparameters belong to the configuration set $(m, n, p, q) \in \mathbb{H} = \{(1, 0, 0, 2), (0, 1, 3, 1), (2, 1, 2, 3), (2, 2, 3, 3), (2, 0, 1, 3)\}$. This set of function includes all the functions with the form*

$$
f(z) = \Phi(w.z^l).z^t, \tag{2}
$$

*with $w, l, t \in \mathbb{R}$ ; $\Phi \in \left\{ \mathrm{id}, \sin, \cos, \sinh, \cosh, \exp, \log(1 + \cdot), \arcsin, \arctan, J_\nu, Y_\nu, I_\nu, \frac{1}{1+\cdot}, \Gamma \right\}$ where $J_\nu, Y_\nu, I_\nu$ are the Bessel functions and $\Gamma$ is Euler's Gamma function.*

*Proof.* Let us use the notation

$$
\tilde{\mathcal{G}}_{p,q}^{m,n} = \left\{ z \mapsto g\left(s.z^r\right) \mid g \in \mathcal{G}_{p,q}^{m,n} \ ; \ s, r \in \mathbb{R} \right\}. \tag{3}
$$

We have to show that the function $f$ appearing in (2) is indeed an element of $\tilde{\mathcal{G}}_{\mathbb{H}}$. We note that Lemmas 1.1 and 1.2 can trivially be extended to the sets $\tilde{\mathcal{G}}$. We now use some tables [2] to check that all the functions appearing in the proposition are included in $\tilde{\mathcal{G}}_{\mathbb{H}}$.

- $id \in \tilde{\mathcal{G}}_{3,1}^{0,1} \subset \tilde{\mathcal{G}}_{\mathbb{H}}$

- $\sin, \cos, \sinh, \cosh \in \tilde{\mathcal{G}}_{0,2}^{1,0} \subset \tilde{\mathcal{G}}_{\mathbb{H}}$

- $\exp \in \tilde{\mathcal{G}}_{0,1}^{1,0} \subset \tilde{\mathcal{G}}_{1,2}^{2,0} \subset \tilde{\mathcal{G}}_{2,3}^{2,1} \subset \tilde{\mathcal{G}}_{\mathbb{H}}$

- $\log(1 + \cdot), \arcsin, \arctan, 1/(1+\cdot) \in \tilde{\mathcal{G}}_{2,2}^{1,2} \subset \tilde{\mathcal{G}}_{3,3}^{2,2} \subset \tilde{\mathcal{G}}_{\mathbb{H}}$

- $J_\nu, I_\nu \in \tilde{\mathcal{G}}_{0,2}^{1,0} \subset \tilde{\mathcal{G}}_{1,3}^{2,0} \subset \tilde{\mathcal{G}}_{\mathbb{H}}$

- $Y_\nu \in \tilde{\mathcal{G}}_{1,3}^{2,0} \subset \tilde{\mathcal{G}}_{\mathbb{H}}$

- $\Gamma \in \tilde{\mathcal{G}}_{1,2}^{2,0} \subset \tilde{\mathcal{G}}_{2,3}^{2,1} \subset \tilde{\mathcal{G}}_{\mathbb{H}}$

We have just showed that any function of the form $f(z) = \Phi(w.z^l)$ is an element of $\tilde{\mathcal{G}}_{\mathbb{H}}$. Therefore, the proposition follows from Lemma 1.2. $\qquad\square$

To speed up the process, the experiments are done by using a restriction of $\mathcal{G}_{\mathbb{H}}$ excluding the inverse trigonometric functions as well as some Bessel functions. This corresponds to $\mathbb{H}' = \{(1,0,0,2), (0,1,3,1), (2,1,2,3)\}$.

## 2 Faithful models go local

In this section, we build a black-box MLP model for the UCI wine quality dataset [4]. We then build a symbolic model for this black-box and we compare our model to a *LIME* explainer [6]. As in the main paper, we split the dataset into a training and a test set. We use the mixup technique [7] on the training set to produce the training data for the symbolic model. Let us start by asking a *LIME* an explanation for an element of the test set. The result is reported on Figure 1.

Figure 1: Feature importance according to a *LIME* predictor for a test prediction. The underlying black-box is a MLP and dataset is the *UCI wine quality* dataset [4].

We change the notation to uppercase letter to match with the notation of *LIME* predictor in Figure 1. It seems like $X_{10}$ is the most important feature for this prediction. A linear expansion of our symbolic model around this test point should give a similar result. Let us produce a first order Taylor expansion of our symbolic model around this point. We obtain the linear local model:

$$
\begin{aligned}
\hat{f}_1(X_0, \dots, X_{10}) = {} & 0.0592X_0 - 0.1217X_1 + 0.3125X_{10} + 0.1015X_2 - 0.05642X_3 - 0.1051X_4 \\
& + 0.0717X_5 - 0.0951X_6 + 0.0473X_7 + 0.1855X_8 + 0.2044X_9 + 0.8399.
\end{aligned}
\tag{4}
$$

We observe that, indeed, $X_{10}$ has the highest weight in this local linear model. Also note that, as suggested by LIME, $X_8$, $X_9$ also have an important weight in this polynomial. However, the agreement is not perfect since $X_5$, $X_6$ have a relatively small weight in our local model. One big advantage of our Symbolic Pursuit scheme is that we can actually choose the order of the local model

obtained by performing a Taylor expansion in a post-hoc analysis. For instance, we can get an idea of nonlinear effects, such as interactions, by going one order above:

$$
\begin{aligned}
\hat{f}_2(X_0, \ldots, X_{10}) =\ & \hat{f}_1(X_0, \ldots, X_{10}) \\
& + 0.0623(-0.0663X_0 + X_1 - 0.7199X_{10} - 0.2145X_2 + 0.2920X_3 + 0.3981X_4 \\
& \qquad - 0.2742X_5 + 0.6176X_6 + 0.6870X_7 + 0.1283X_8 - 0.5097X_9 - 0.7128)^2 \\
& - 0.0002(0.1332X_0 + 0.3422X_1 + 0.4388X_{10} + 0.1563X_2 + 0.0363X_3 - 0.0364X_4 \\
& \qquad + 0.0231X_5 + 0.1504X_6 + 0.6342X_7 + 0.6567X_8 + 0.2593X_9 - 1)^2.
\end{aligned}
\tag{5}
$$

We note that by introducing the new variables

$$
\begin{aligned}
z_1 =\ & -0.0663X_0 + X_1 - 0.7199X_{10} - 0.2145X_2 + 0.2920X_3 + 0.3981X_4 \\
& - 0.2742X_5 + 0.6176X_6 + 0.6870X_7 + 0.1283X_8 - 0.5097X_9 - 0.7128
\end{aligned}
\tag{6}
$$

$$
\begin{aligned}
z_2 =\ & 0.1332X_0 + 0.3422X_1 + 0.4388X_{10} + 0.1563X_2 + 0.0363X_3 - 0.0364X_4 \\
& + 0.0231X_5 + 0.1504X_6 + 0.6342X_7 + 0.6567X_8 + 0.2593X_9 - 1,
\end{aligned}
$$

and the coefficients $c_0 = 1,001$, $c_{1,1} = -0.2339$, $c_{1,2} = 0.3280$, $c_{2,1} = 0.0623$, $c_{2,2} = -0.0002$, the second order Taylor expansions can be rewritten very easily in the compact form:

$$
\begin{aligned}
\hat{f}_1(z_1, z_2) &= c_0 + c_{1,1} \cdot z_1 + c_{1,2} \cdot z_2 \\
\hat{f}_2(z_1, z_2) &= \hat{f}_1(z_1, z_2) + c_{2,1} \cdot (z_1)^2 + c_{2,2} \cdot (z_2)^2
\end{aligned}
\tag{7}
$$

In this way, the user can easily identify the new variables among the $z$'s that contribute the most by simply inspecting the coefficients $b$'s. For each new variable $z$, the user can identify the features that contribute with the highest weight in the affine combination encoded in each $z$. Therefore, the local models produced by our symbolic pursuit model allow the users to easily spot the most important features and the most important interactions. How is that so? This is in fact a simple consequence of the fact that we use a projection pursuit algorithm. Recall that our symbolic model was of the form:

$$
\hat{f}(x) = \sum_{k=1}^{K} w_k \cdot g_k \left( \left[ \frac{v_k^\top x}{\|v_k\|\sqrt{d}} \right]^+ \right).
\tag{8}
$$

Let $x_0 \in \mathcal{X}$ be a feature point such that $\forall k \in K : v_k^\top x_0 \neq 0$. We can write a Taylor expansion of order $N$ for this model for $x$ around this point very easily:

$$
\hat{f}_n(x) = \hat{f}(x_0) + \sum_{k=1}^{K} \sum_{n=1}^{N} \frac{1}{n!} w_k \cdot g_k^{(n)} \left( \left[ \frac{v_k^\top x_0}{\|v_k\|\sqrt{d}} \right]^+ \right) \cdot \left( \left[ \frac{v_k^\top (x - x_0)}{\|v_k\|\sqrt{d}} \right]^+ \right)^n,
\tag{9}
$$

where $g^{(n)}$ denotes the $n$-th derivative of $g$. By introducing new variables

$$
z_k = \frac{v_k^\top (x - x_0)}{\|v_k\|\sqrt{d}}, \qquad\qquad \forall k \in \{1, \ldots, K\},
\tag{10}
$$

we can, as above, rewrite the Taylor expansion in a very compact form:

$$
\hat{f}_n(z_1, \ldots, z_K) = \hat{f}(x_0) + \sum_{k=1}^{K} \sum_{n=1}^{N} \frac{1}{n!} w_k \cdot g_k^{(n)} \left( \left[ \frac{v_k^\top x_0}{\|v_k\|\sqrt{d}} \right]^+ \right) \cdot \left( [z_k]^+ \right)^n.
\tag{11}
$$

This is Taylor expansion has precisely the same form as (7) for $K = 2$, $N = 2$. We finish on a last remark on the benefits offered by our projection pursuit approach. We see that both the symbolic model and its local approximation take a very concise form when we consider the new variables $z_k$, $k = 1, \ldots, K$. This is another advantage of our approach: we discover new variables, which are affine combination of the features, whom with the discussion simplifies.

## 3  Irregularity of the loss surface

In this Section, we show that the loss surfaces admit singular regions as a consequence of Meijer G-functions properties. For simplicity, we restrict to the case of a single feature $x \in \mathcal{X} = [0, 1]$ so that we approximate the Black-box model $f$ with a single Meijer G-function. It goes without saying that this discussion extends naturally to the case of multiple features. We also assume that neither $f$ nor the estimator is singular at $x = 0$ to guarantee that the singularity emerges from the properties of Meijer G-functions. The following result indicates some general conditions under which the loss surface is singular:

**Proposition 3.1.** *Let $f : [0, 1] \to \mathbb{R}$ be a continuous function such that $f(x) \sim x^u$ for $x \to 0^+$, where $u \in \mathbb{R}^+$. We approximate this function by a Meijer G-function in the set $\mathcal{G}_{p,q}^{m,n}$ so that the MSE loss is given by*

$$L(\boldsymbol{a}, \boldsymbol{b}) = \int_0^1 dx \left( f(x) - G_{p,q}^{m,n} \left( \left. \begin{array}{c} \boldsymbol{a} \\ \boldsymbol{b} \end{array} \right| x \right) \right), \tag{12}$$

*where $\boldsymbol{a} \in \mathbb{R}^p$ and $\boldsymbol{b} \in \mathbb{R}^q$ . Suppose that this loss is minimized with a Meijer G-function*

$$\hat{f}(x) = G_{p,q}^{m,n} \left( \left. \begin{array}{c} \boldsymbol{a}^* \\ \boldsymbol{b}^* \end{array} \right| x \right) \sim x^v \qquad \text{for } x \to 0^+, \tag{13}$$

*where $v \in \mathbb{R}^+$. Define the half-line through the optimum*

$$\mathcal{D} = \left\{ (\boldsymbol{a}^* + \lambda \mathbf{1}_p , \, \boldsymbol{b}^* + \lambda \mathbf{1}_q) \, \middle| \, \lambda < \max \left( -\frac{1 + 2v}{2}, -(1 + u + v) \right) \right\} \subset \mathbb{R}^{p+q}. \tag{14}$$

*Then the MSE loss (12) diverges on $\mathcal{D}$.*

*Proof.* The proof is a simple application of the following property of Meijer G-functions:

$$G_{p,q}^{m,n} \left( \left. \begin{array}{c} \boldsymbol{a} + \lambda \mathbf{1}_p \\ \boldsymbol{b} + \lambda \mathbf{1}_q \end{array} \right| x \right) = x^\lambda \, G_{p,q}^{m,n} \left( \left. \begin{array}{c} \boldsymbol{a} \\ \boldsymbol{b} \end{array} \right| x \right) \quad \lambda \in \mathbb{R}. \tag{15}$$

We use this to rewrite the loss around the optimum as

$$\begin{aligned} L(\boldsymbol{a}^* + \lambda \mathbf{1}_p, \boldsymbol{b}^* + \lambda \mathbf{1}_q) &= \int_0^1 dx \left[ f(x) - G_{p,q}^{m,n} \left( \left. \begin{array}{c} \boldsymbol{a}^* + \lambda \mathbf{1}_p \\ \boldsymbol{b}^* + \lambda \mathbf{1}_q \end{array} \right| x \right) \right]^2 \\ &= \int_0^1 dx \left[ f(x) - x^\lambda \, G_{p,q}^{m,n} \left( \left. \begin{array}{c} \boldsymbol{a}^* \\ \boldsymbol{b}^* \end{array} \right| x \right) \right]^2. \end{aligned} \tag{16}$$

We note that the integrand behaves as

$$\left[ f(x) - x^\lambda \, G_{p,q}^{m,n} \left( \left. \begin{array}{c} \boldsymbol{a}^* \\ \boldsymbol{b}^* \end{array} \right| x \right) \right]^2 \sim x^{2u} + x^{2v+2\lambda} - 2.x^{u+v+\lambda} \qquad \text{for } x \to 0^+. \tag{17}$$

It follows trivially that the integral in (16) diverges whenever $2v + 2\lambda < -1$ or $u + v + \lambda < -1$, which proves the proposition. $\qquad \square$

This proposition indicates a region of the parameter space where the loss is ill-defined. The first thing that should be noted is that the loss might be ill-defined on a larger region than $\mathcal{D}$ under more restrictive assumptions.

To make this proposition more explicit, we give a simple example where this proposition applies. Assume that $f(x) = \exp(-x)$. It follows immediately that $u = 0$. As a matter of fact, this function can be perfectly approximated with the following Meijer G-function:

$$\hat{f}(x) := G_{0,1}^{1,0} \left( \left. \begin{array}{c} \overline{\phantom{x}} \\ 0 \end{array} \right| x \right) = \exp(-x). \tag{18}$$

Figure 2: True loss compared to the numerical approximation computed with the library `mpmath`.

As above, it follows that $v = 0$. Any Meijer G-functions in $\mathcal{G}_{0,1}^{1,0}$ is parametrized by a single real parameter $b \in \mathbb{R}$ and can be written

$$f_b(x) = G_{0,1}^{1,0} \left( \begin{array}{c} - \\ b \end{array} \middle| \; x \; \right). \tag{19}$$

If we apply Proposition 3.1, we find that the loss diverges whenever $b < -1/2$. In this case though, we can evaluate the loss explicitly:

$$
\begin{aligned}
L(b) &= \int_0^1 dx \left( \exp\left(-x\right) - G_{0,1}^{1,0} \left( \begin{array}{c} - \\ b \end{array} \middle| \; x \; \right) \right)^2 \\
&= \int_0^1 dx \; \exp\left(-2x\right) \left(1 - x^b\right)^2 .
\end{aligned}
\tag{20}
$$

From (20), we see that when $b < 0$, the integrand behaves as $x^{-b}$ when $x \to 0^+$ so that the integral diverges whenever $b < -1/2$. Everything is consistent with Proposition 3.1.

It is instructive to compare the analytic result (20) to a numerical evaluation of the loss using the `meijerg` class from the Python library `mpmath` [5]. We note on Figure 2 that the numerical approximation computed with `mpmath` cannot be trusted as we are getting closer to the singularities $b < -1/2$. This figure confirms that the numerical loss surface is accurate above the optimum $b = 0$. Therefore, we have to approach the optimum by above ($b > 0$ here) to avoid the singularities.

By giving a closer look at Figure 2, we can make another remark about the loss landscape of Meijer G-function approximations. We note that the loss surface is flat near the optimum $b = 0$. In this setup, this simply tells us that many Meijer G-function can approximate the Black-box function $f$ with a very good accuracy. In practice, there is no guarantee that all these Meijer G-functions in the vicinity of the optimum reduce to the same functional form as the optimum. For instance, a bad case scenario would be the one depicted in Figure 3: most of the Meijer G-function around the optimum in the parameter space don't reduce to a function whose expression looks like the optimum. To relate this to our above example, this would translate into Meijer G-function that are not exponential when $b$ is not close enough to zero. However, in practice, if we have a symbolic model that is accurate and whose expression can realistically be digested by a human brain, our symbolic model already provides a sensible way to probe the black-box.

Figure 3: A possible bad case scenario in the parameter space of a single Meijer G-function.

## 4   Cancelling poles with a lasso penalty

As we have seen in Section 1, it is possible to cancel some poles in the integral defining the Meijer G-function. Let us take a simple example to see how it works. We start with a function included in $\mathcal{G}_{2,2}^{1,1}$ and show that it reduces to a function included in $\mathcal{G}_{1,1}^{0,1}$:

$$
\begin{aligned}
G_{2,2}^{1,1} \left( \begin{array}{c} 1,\pi \\ \pi,2 \end{array} \middle| x \right) &= \frac{1}{2\pi i} \int_{\mathcal{L}} ds\, x^s \frac{\Gamma(\pi - s)\Gamma(1-1+s)}{\Gamma(1-2+s)\Gamma(\pi - s)} \\
&= \frac{1}{2\pi i} \int_{\mathcal{L}} ds\, x^s \frac{\Gamma(1-1+s)}{\Gamma(1-2+s)} \\
&= G_{1,1}^{0,1} \left( \begin{array}{c} 1 \\ 2 \end{array} \middle| x \right) \in \mathcal{G}_{1,1}^{0,1}.
\end{aligned}
\tag{21}
$$

More generally, we will now change slightly the notation for Meijer G-function to make the parts that can cancel with each other more explicit. Let us denote the Meijer G-function in the following way:

$$
G_{p,q}^{m,n} \left( \begin{array}{c} \boldsymbol{\alpha},\boldsymbol{\gamma} \\ \boldsymbol{\beta},\boldsymbol{\delta} \end{array} \middle| x \right),
\tag{22}
$$

where $\boldsymbol{\alpha} = (a_1,\ldots,a_n)$ ; $\boldsymbol{\gamma} = (a_{n+1},\ldots,a_p)$ ; $\boldsymbol{\beta} = (b_1,\ldots,b_m)$ ; $\boldsymbol{\delta} = (b_{m+1},\ldots,b_q)$. By comparing with the above example, we see that the elements of $\boldsymbol{\alpha}$ can cancel the elements of $\boldsymbol{\delta}$ while the elements of $\boldsymbol{\beta}$ can cancel the elements of $\boldsymbol{\gamma}$. How can we enforce this poles to simplify in order to end up with a simpler Meijer G-function? This can be done by adding a Lasso term to the MSE loss that we define in Section 4 of the paper. Remember that our Symbolic Pursuit algorithm solves the following itteration problem at each iteration:

$$
(g_k, v_k, w_k) = \arg \min_{\mathcal{G}_{\mathbb{H}} \times \mathbb{R}^d \times \mathbb{R}} \int_{\mathcal{X}} dx \left[ r_k(x) - w.g \left( \left[ \frac{v^T \cdot x}{||v||\sqrt{d}} \right]^+ \right) \right]^2.
\tag{23}
$$

In each subset $\mathcal{G}_{p,q}^{m,n}$ of $\mathcal{G}_{\mathbb{H}}$, this corresponds to the following loss with respect to the residual $r$:

$$
L(\boldsymbol{\alpha},\boldsymbol{\beta},\boldsymbol{\gamma},\boldsymbol{\delta},v,w) = \int_{\mathcal{X}} dx \left[ r(x) - w.G_{p,q}^{m,n} \left( \begin{array}{c} \boldsymbol{\alpha},\boldsymbol{\gamma} \\ \boldsymbol{\beta},\boldsymbol{\delta} \end{array} \middle| \left[ \frac{v^T \cdot x}{||v||\sqrt{d}} \right]^+ \right) \right]^2.
\tag{24}
$$

We could add to this loss a penalty that pushes poles to cancel with zeroes very simply:

$$
\begin{aligned}
\tilde{L}(\boldsymbol{\alpha},\boldsymbol{\beta},\boldsymbol{\gamma},\boldsymbol{\delta},v,w) = {}& L(\boldsymbol{\alpha},\boldsymbol{\beta},\boldsymbol{\gamma},\boldsymbol{\delta},v,w) + \frac{\lambda}{\min(p,q-m)} \sum_{i=1}^{\min(p,q-m)} |\alpha_i - \delta_i| \\
& + \frac{\lambda}{\min(q,p-n)} \sum_{i=1}^{\min(q,p-n)} |\beta_i - \gamma_i|,
\end{aligned}
\tag{25}
$$

with the convention that $^1/_0 \sum_{i=1}^0 (\ldots) = 0$ and $\lambda \in \mathbb{R}$ being the Lasso coefficient. Note that we have normalized each sum by its number of terms so that the Meijer G-function that contain more zero-pole pairs are not penalized. This new optimization problem is naturally endowed with a threshold : if one of the term inside the 4 sums in (25) is bellow the coefficient $\lambda$, this term is set to zero so that the cancellation mechanism occurs. Finally, we stress that this new optimization problem should come in complement to the optimization that is solely based on the MSE. Because we loose accuracy with the lasso, it should be used in the case where the model obtained with the MSE only is not interpretable. In the worst case scenario where neither the MSE nor the lasso optimization produce an interpretable result, we could try a search algorithm around the optimum to look for an interpretable approximation. We leave this track open for future works.

# 5 Pseudo code of Symbolic Pursuit

The pseudocode that we use in our implementation of the Symbolic Pursuit algorithm is the following.[2]

---
**Algorithm 1:** Symbolic Pursuit

---
**Input** : Black-box Model $f$ ; Training Set $\mathbb{T} = \{x_1, \ldots, x_N\} \subset \mathcal{X}$
**Output :** Symbolic model $\hat{f}$ for $f$
Draw $N$ weights $\lambda_1, \ldots, \lambda_N \sim \text{Beta}\,(0.2, 0.2)$ ;
Draw $N$ couples $(\tilde{x}_1^A, \tilde{x}_1^B), \ldots, (\tilde{x}_N^A, \tilde{x}_N^B) \sim U\left(\mathbb{T}^2\right)$ ;
Initialize the symbolic training set $\tilde{x}_i = \lambda_i \tilde{x}_i^A + (1 - \lambda_i)\tilde{x}_i^B, \;\; \forall i = 1, \ldots, N$ ;
Initialize the residual vectors $r_0 \leftarrow [f(\tilde{x}_i)]_{i=1,\ldots,N}$ and $r_1 \leftarrow 0$ ;
**while** $\|r_{k+1}\|/\|r_k\| < $ *tolerance* **do**

$\quad (g_k, v_k, w_k) \leftarrow \arg\min_{\mathcal{G}_{\mathbb{H}} \times \mathbb{R}^d \times \mathbb{R}} \sum_{i=1}^N \left[ r_k(\tilde{x}_i) - wg\left( \left[ \frac{v^\top \tilde{x}_i}{\|v\|\sqrt{d}} \right]^+ \right) \right]^2$ ;

$\quad r_{k+1} \leftarrow r_k - \left[ w_k.g_k\left( \left[ \frac{v_k^\top \tilde{x}_i}{\|v_k\|\sqrt{d}} \right]^+ \right) \right]_{i=1,\ldots,N}$ ;

$\quad$ **for** $l = 1, \ldots, k-1$ **do**

$\quad\quad (m, n, , p, q) \leftarrow \text{hyperparameters}(g_l)$;

$\quad\quad (g_l, v_l, w_l) \leftarrow \arg\min_{\mathcal{G}_{p,q}^{m,n} \times \mathbb{R}^d \times \mathbb{R}} \sum_{i=1}^N \left[ r_{k,l}(\tilde{x}_i) - wg\left( \left[ \frac{v^\top \tilde{x}_i}{\|v\|\sqrt{d}} \right]^+ \right) \right]^2$ ;

$\quad\quad r_{k+1} \leftarrow r_k - \left[ w_l.g_l\left( \left[ \frac{v_l^\top \tilde{x}_i}{\|v_l\|\sqrt{d}} \right]^+ \right) \right]_{i=1,\ldots,N}$ ;

$\quad$ **end**

$\quad k \leftarrow k + 1$ ;

**end**

$\hat{f}(x) \leftarrow \sum_k w_k g_k\left( \left[ \frac{v_k^\top \cdot x}{\|v_k\|\sqrt{d}} \right]^+ \right)$ ;

---

Note that, to alleviate the back-fitting procedure, we have restricted the optimization correcting each term to the subset $\mathcal{G}_{p,q}^{m,n}$ of $\mathcal{G}_{\mathbb{H}}$. Keep in mind that the black box model we are trying to interpret has been trained on some particular dataset, and that different training data would probably yield a different black box model. On the one hand, we do not want to train our interpreter on the same training data as the black box model because that would tend to produce another predictive model rather than an interpretation of the black box model. On the other hand, we do not want to train our interpreter on some data that is unconnected to the data that was used to train the black box model. We therefore use a *mixup* strategy [7]: we create new samples whose features $x$ are random convex combinations of the features of samples in the actual training data and query the black box model for the value $f(x)$.

# 6 Experiments

In the our experiments with the real-world datasets, we use the usual definitions for the MSE and the determination coefficient $R^2$. However, there is a slight subtlety due to the fact that we are comparing several quantities. To make this clear, let us assume that we have a black-box $f : \mathcal{X} \to \mathcal{Y}$ and the associated symbolic model $\hat{f} : \mathcal{X} \to \mathcal{Y}$. Let $\{(x_i, y_i) \in \mathcal{X} \times \mathcal{Y} \mid i = 1, \ldots, N\}$ be a test set used to evaluate the performance of our models. Let us now give a precise definition of the metrics appearing in the main paper in all the possible scenarios. When we are evaluating the performances of the black-box, we use the following metrics:

$$MSE = \frac{1}{N} \sum_{i=1}^{N} (y_i - f(x_i))^2 \tag{26}$$

$$R^2 = 1 - \frac{\sum_{i=1}^{N} (y_i - f(x_i))^2}{\sum_{i=1}^{N} (y_i - \bar{y})^2}, \tag{27}$$

where $\bar{y} = N^{-1} \sum_{i=1}^{N} y_i$ denotes the average of the test labels. These two metrics correspond to the column Black-Box in Table 2 from the main paper. We can easily adapt the definition of these metrics to asses the performances of the symbolic model:

$$MSE = \frac{1}{N} \sum_{i=1}^{N} \left( y_i - \hat{f}(x_i) \right)^2 \tag{28}$$

$$R^2 = 1 - \frac{\sum_{i=1}^{N} \left( y_i - \hat{f}(x_i) \right)^2}{\sum_{i=1}^{N} (y_i - \bar{y})^2}, \tag{29}$$

These two metrics correspond to the column Symbolic in Table 2 from the main paper. Finally, we can adapt the definition of these metrics to asses the quality of $\hat{f}$ as an approximation of $f$:

$$MSE = \frac{1}{N} \sum_{i=1}^{N} \left( f(x_i) - \hat{f}(x_i) \right)^2 \tag{30}$$

$$R^2 = 1 - \frac{\sum_{i=1}^{N} \left( f(x_i) - \hat{f}(x_i) \right)^2}{\sum_{i=1}^{N} \left( f(x_i) - \bar{f} \right)^2}, \tag{31}$$

where $\bar{f} = N^{-1} \sum_{i=1}^{N} f(x_i)$ denotes the average of the black-box labels. These two metrics correspond to the column Symbolic vs Black-Box in Table 2 from the main paper.

## Footnotes

[1]The poles are associated to the Gamma functions appearing in the numerator of the integrand. Details can be found in [3].

[2]In this section and in Section 6, the index $i$ in $x_i$ is used to label different samples from a set of data points. This is different from the notations in the main paper and from Section 2 where this index is used to label the components of a given data point. We did this slight change so that the notations remain concise and familiar.