[Reviews · NeurIPS 2020]

Review 1

Summary and Contributions: The paper proposes a method to globally approximate any continuous black-box function using Meijer G-functions instead of polynomial splines. The method is tested on an SVM and MLP trained on 5 UCI datasets which contain attribute data.

Strengths: - The paper demonstrates that the method achieves good results on the tested domain and black-box models; - The method is compared against relevant alternatives; - The method succeeds in achieving a parsemonious representation, satisfying important XAI model requirements

Weaknesses: Model agnostic explanation methods find strength in the fact they can be used on various types of models and implicitly datasets, often including text and images, see [1, 2] yet, in this paper only datasets from UCI repository is used and this type of data is not representative of the wide range of datatypes that could be used. In itself, this is not problematic, however the paper would be significantly improved if there was some indication of how well this method would translate outside of the tested domain. For example what can I expect if I wanted to approximate a modern CNN trained on an image classification dataset? [1] Baehrens, D., Schroeter, T., Harmeling, S., Kawanabe, M., Hansen, K., & MÞller, K.-R. (2010). How to explain individual classification decisions. Journal of Machine Learning Research, 11 (Jun), 1803–1831. [2] Ribeiro, M. T., Singh, S., & Guestrin, C. (2018). Anchors: High-precision model-agnostic explanations. In Thirty-Second AAAI Conference on Artificial Intelligence.

Correctness: I think so

Clarity: The paper is well written from a grammatical, style, and organizational point of view. The proposed method and novel concepts are well articulated. Section names and titles are well done, and contain pertinent pieces of information. So well presented from content and style viewpoint.

Relation to Prior Work: Yes

Reproducibility: No

Additional Feedback: Post-rebuttal comment: I thank the authors for their feedback. The method achieves good results on the tested domain and black-box models and the authors indicate why their method has advantages over symbolic metamodels.


Review 2

Summary and Contributions: I read the feedback from the authors. I satisfy their answers to my questions. I keep the same position for accepting this paper. I consider the key advantage of this work is showing a good direction in terms of functional level discovery, which is mostly critical in explainable AI. This papers presented a study of projection pursuit using Meijer G-functions, which was able to produces analytical interpretations about the black-box model. The authors called their algorithm to be Faithful Pursuit. They conducted numerical studies on two popular black-box models, Multilayer Perceptron (MLP) and Support Vector Machine (SVM), from five UCI datasets. They compared the performance of Faithful Pursuit with the two black-box models. For the interpretable property of Feature Importance, they showed one example in comparison with a local linear surrogate model (LIME). The proposed Faithful Pursuit was able to capture nonlinear properties for interpretation which is beyond the capacity of a linear model.

Strengths: I like this study from its principle of constructing interpretable models based on Meijer G-functions. The study was in the same principle of [1], but provided a further development. First, the authors investigated the problem from the view of Projection Pursuit. Second, they derive the tree relations between the hyperparameters of Meijer G-functions and the most familiar functions. Third, their model showed the properties of parsimonious with a small number of free parameters and efficiency from using differentiable forms directly in learning. Fourth, they used a mixup strategy in training the interpreter which is meaningful. I consider the study is solid and shows the significance and novelty of the contribution. The study will be interested to most researchers in the NeurIPS community. I agree with the point by the authors about "a new and very promising direction" for this study. It shows a direction for a functional form learning, which is the most important information to understand the system investigated.

Weaknesses: I consider this study is similar to [1] in the principle in constructing models based on Meijer G-functions. In [1], they called their model "Symbolic Metamodels". In this study, the authors called their model "Faithful Pursuit", or " faithful model". For me, it is ok to give a new name, but the main difference between two models seems to be on the parsimonious property, instead of faithful one. I do not think "faithful" is a suitable term to describe the new model, which may be misleaded or confused to the users.

Correctness: I consider the study is in generally correct on the claims and methods.

Clarity: Yes, in most parts. However, I have difficulties in some parts given below. 1. I have a difficulty to understand #Terms in Table 2 and its explanations. For me, the solution for the Faithful model should output the selected (m,n,p,q) from H. which is an important information about the functional forms learned. 2. In the Supplementary Material, I cannot see why H = (0,1,3,1) is selected? It is for approximating what types of functions. 3. I have a difficulty to understand Figure 3 in the Supplementary Material and its explanations. Did the authors want to say the other Meijer G-functions are able to get the similar solution around the optimum so that it is difficult to tell which function should be selected.

Relation to Prior Work: Yes, in a general sense.

Reproducibility: Yes

Additional Feedback: 4. If the authors would like to keep the term "faithful", it is better to define it. I know the work by Andrews, R., Diederich, J., & Tickle, A.B. (1995). Survey and critique of techniques for extracting rules from trained artificial neural networks. Knowledge-based Syst., 8, 6, 373-389. where they used the term "fidelity" for the similar study. This is why to define the term. 5. What did authors mean for "a global interpretation"? If a model is faithful to the physical system, it should be able to cover the locality property (higher orders of Taylor expansions?). I was confused on this point for the reason of missing definitions. 6. The data of the selected (m,n,p,q) is better to be given, such as from one example in Table 2. Readers will know the selected functional forms in the different k.


Review 3

Summary and Contributions: The authors propose a method for approximating a given continuous black-box function with a linear combination of (a subclass of) Meijer G-functions in an attempt to generate its ``global interpretation''. In particular, they identified a small subset of the Meijer G-functions that already covers a large class of many familiar functions (e.g., polynomial, rational, exponential, trigonometric and hypergeometric functions) and empirically showed that such subset suffices to accurately approximate supervised models (SVM and MLP) trained on five UCI datasets. The generated approximation is global in the sense that it approximates the original black box function on the whole input space, and is more interpretable in the sense that it is a sum of functions of well-known forms. While the evaluation remains rather limited, the authors demonstrate a gain from their method in the interpretability aspect by peforming a 2nd order Taylor expansion of the approximation and showing on the Wine dataset that interaction between some features may play an important role in prediction --- a form of information that cannot be provided by the prior popular interpretability methods such as DeepLift, SHAP, LIME, etc that can only quantify importance of individial features independently.

Strengths: - the idea of employing the combination of projection persuit and back-fitting to iteratively approximate a black-box function with a linear combination of G-functions seems intuitive and works well on some relatively low-dimensional public datasets. The identification of an expressive subset circumvents the need for hyper-parameter optimization.

Weaknesses: - The validation of the interpretability aspect of the proposed method seems rather preliminary. Quantitative validations or comparison with other relevant methods are absent. For example, evaluating the feature importace values in synthetic experiments where the underlying true feature ranking is known could be added e.g., Fig.4 in [1] might be a good option. Since the proposed method can also capture feature interaction, designing a similar synthetic scenario where the importance of pairwise interactions are known might also be worthwhile. - The advantages of the proposed method over the most relevant prior work [1] still remains unclear. It would be nice if the authors could include an example or an experiment in which some aspects of the two methods are compared in concrete terms. For example, a comparison of the two methods in the prognosis prediction task for breast cancer (as done in [1]) may be informative. [1] demystifying black-box models with symbolic metamodels, NeurIPS 2019

Correctness: I believe that the notion of interpretability should be defined more precisely in the specific context of this work. If a "non-interpretable" blackbox function is approximated by a linear combination of (potentiall many) functions of well-known forms (e.g., polynomial, rational, exponential, trigonometric and hypergeometric functions), what kinds of interpretations could we elicit that we otherwise could not? I believe the Taylor approximation to capture higher order interactions is one form of such interpretations, but there could be more to this, and a more extended discussion would be a valuable addition.

Clarity: The paper is well-written overall, but I believe that the paper would benefit a lot from concretising the specific notion of interpretability that they aim to address in their work (see my comments to the above section).

Relation to Prior Work: While the conceptual differences w.r.t. the closest prior work [1] have been clearly discussed in the related work section, no experiments have been performed to demonstrate convincingly how these differences would matter in practice. Alternatively, perhaps a toy-ish example where such differences would affect the interpretability of the resultant approximations could be included. [1] demystifying black-box models with symbolic metamodels, NeurIPS 2019

Reproducibility: Yes

Additional Feedback: Minor comments: - line 106: the authors describe that projection persuit with polynomial splines is less interpretable because the indicidual terms (polynomials) may not have natural interpretations ... what do you mean by this? - line 212: "in" => "is" - line 249: R^2 is not defined -------------------------- after rebuttal --------------------------------- Having read the author response and other reviews, I have revised my score from 5 to 6. This was my attempt to calibrate my score based on the reviews of the other reviewers, and the authors have provided extra evidence on the benefits of the proposed method. However, I still remain unchanged in my view that the quality of empirical comparison of the gained interpretability is insufficient to claim practical improvements.


Review 4

Summary and Contributions: The paper proposes an algorithm, based on the use of Meijer G-functions, that produces global interpretation of any given continuous black-box function. In this way, the topic of interpreting Machine Learning models is once again covered, proposing a method that provides, as other methods also do, knowledge about feature importance and feature interaction, as well as independence to the model used. However, it differs from most of the other existing methods precisely in that it provides global information and not only local information. The last aspect that makes this algorithm remarkable is that it is an algorithm that produces parsimonious expressions.

Strengths: The claims in this article are sound. They are based on a strong mathematical basis and are supported by experimentation The topic discussed (interpretability in Machine Learning models) is a topic of high interest. The proposal made is adequately compared with previous techniques and shows better properties

Weaknesses: This is a work that shows quite a lot of mathematical complexity. Thus, the jumps along the article necessary to understand some parts of it make it difficult to understand at certain times. It is a work that represents the first step in a new promising line so that the experimentation performed was not done with the use of datasets or complex models.

Correctness: - Claims, methods, and empirical methodology are correct.

Clarity: The paper is well written. In order to help in some corrections, it could be nice to note some minor faults: * Sometimes to express multiplication, instead of using the appropriate point, a normal point is used (the appropriate point corresponding to the multiplication is centered and not at the bottom). See formula 12 (bad) vs formula 13 (well) as an example. * Sometimes the term backfitting is used and sometimes back-fitting. Unify this. * In section 6, line 279, figure 2 b refers to v, not to v1. It is imperative to clarify again that the paper is well written and errors noted are only to help future corrections.

Relation to Prior Work: Works in the same direction are shown and are compared by a comparative table of properties of these previous works and the proposed work, showing the advantages that the latter has over such previous works.

Reproducibility: Yes

Additional Feedback: Having read the author response and other reviews, I decided to keep my previous score. I think it is a good paper which deserves an acceptation.

[Author Response · NeurIPS 2020]

We thank all the reviewers for excellent questions and many relevant remarks.

**[[Reviewer 1]]** ■ **On extensions to other domains (e.g. CNN):** Thank you for this remark. While our method can in principle be used for various datasets and black-box architectures, we are focusing on tabular datasets as it is the case in [1]. Media-specific tasks, such as image classification and natural language processing, are beyond the scope of this paper. One of the reason for this is that our method produces interpretations directly in terms of the input features. In the case of CNN, for instance, we believe that meaningful interpretations should involve hidden units explicitly, see [2] for instance.

**[[Reviewer 2]]** ■ **Usage of the word faithful:** Thank you for pointing this out, we agree that faithful is not best. As you point out, our models are indeed symbolic models; for this reason, we will rename our algorithm Symbolic Pursuit in the final manuscript. ■ **Hyperparameters in Table 2:** We did not include these details in Table 2 for the sake of conciseness. We simply wanted to demonstrate that the interpretable models we obtained are accurate (have high $R^2$ scores) and concise (have a small number of terms). We shall report the details in the Supplementary Material of the final manuscript. ■ **Interpretation of Figure 3:** The explanation provided for Figure 3 from the Supplementary Material in the review is correct; we will make this more explicit in the final manuscript. ■ **Meaning of global interpretations:** In the context of this work, we mean that our models are good global approximations of the black-boxes (see the high $R^2$ scores in Table 2 ). This is not the case for local models such as LIME.

**[[Reviewer 3]]** ■ **Quantitative comparison with other methods:** Many thanks for this suggestion; we will include such experiments in the final manuscript. In the mean time, we ran a toy experiment: We drew 100 inputs $(x_1, x_2) \sim U\left([0,1]^2\right)$ and set, as a pseudo black box function, $f(x_1, x_2) = 0.3 \cdot x_1 + 0.6 \cdot x_2$. In this simple setup, the feature $x_2$ is everywhere twice as important than the feature $x_1$; this corresponds to having a normalized feature importance vector everywhere equal to $(0.44, 0.88)$. We used our method and two others to produce feature importance vectors at four test points; the results are reported in Table 1. We see that our method produces feature importance vectors that are very similar and close to the actual vectors; the other methods produce feature importance vectors that are very different from the actual vectors and very different at the various test points. We will add more sophisticated experiments in the final manuscript.

■ **Advantages over Symbolic Metamodels [1]:** There are two main advantages. The first one, as discussed in lines 221-226, is that our method produces expressions with many fewer terms than those produced by Symbolic Metamodels. For instance, on the experiments that are synthesized in Table 2, a Symbolic Metamodel would need 78 terms for Wine, 28 for Yacht, 105 for Boston, 36 for Energy, 45 for Concrete; we think that interpreting an expression with so many terms would be very difficult. By contrast (see Table 2), for these

Table 1: Normalized feature importance vector.

| Test point | Our Method | LIME | SHAP |
|---|---|---|---|
| 1 | $(0.44, 0.92)$ | $(0.84, 0.52)$ | $(0.89, 0.45)$ |
| 2 | $(0.42, 0.90)$ | $(0.20, 0.97)$ | $(0.28, 0.95)$ |
| 3 | $(0.44, 0.92)$ | $(0.85, 0.51)$ | $(0.49, 0.86)$ |
| 4 | $(0.44, 0.92)$ | $(0.55, 0.83)$ | $(0.70, 0.70)$ |

datasets our method almost always produces models with only 1, 2 or 3 terms. The second advantage is that we address hyperparameter selection (see Section 3). This allows more flexibility than Symbolic Metamodels in the function(s) that we are able to identify (see especially lines 226-232). The comparison for the breast cancer dataset is unfortunately impossible since the dataset is not public. ■ **Precision on the notion of interpretability:** We believe that our method helps interpretability by outputting a model whose mathematical expression can realistically be written and manipulated by a human subject (which is obviously not the case for the black-box). An illustration of this, as mentioned in the review, is the possibility of building a Taylor approximation to extract linear effects (such as feature importance) and non-linear effects (such as feature interactions). We also suggest that various linear combinations of the features appearing in the model at Equ. (10) can be interpreted as new variables in which the model takes a simple form. This offers additional conciseness (on top of the parsimonious size of the model), which facilitates the analysis by a human subject. ■ **On the interpretability of polynomial splines:** Thank you for this remark, we shall clarify in the final manuscript. We would like to avoid restricting to polynomials for the reasons explained in line 28-46, but it is more a matter of representation power rather than interpretation. ■ **On the definition of $R^2$:** By $R^2$ we mean the usual coefficient of determination; we will give a precise definition in the revision. Thank you for pointing this out.

**[[Reviewer 4]]** ■ **Notational inconsistencies:** Many thanks for pointing out the notational inconsistencies; we will correct them in the final manuscript.

[1] Ahmed M. Alaa and Mihaela van der Schaar. Demystifying black-box models with symbolic metamodels. In H. Wallach, H. Larochelle, A. Beygelzimer, F. dAlché-Buc, E. Fox, and R. Garnett, editors, *Advances in Neural Information Processing Systems 32*, pages 11304–11314. Curran Associates, Inc., 2019.

[2] David Bau, Bolei Zhou, Aditya Khosla, Aude Oliva, and Antonio Torralba. Network dissection: Quantifying interpretability of deep visual representations. *2017 IEEE Conference on Computer Vision and Pattern Recognition*


[Meta-Review · NeurIPS 2020]

All referees support acceptance for the contributions, notably R#2 and R#4, and I also recommend acceptance after reading the reviews, rebuttal and discussions. However, please consider revising your paper to address reviewer R#3’s & R#4’s remark on typos and R#2’s comments on "faithful".